# Assessing Smoke-Free Housing Implementation Approaches to Inform Best Practices: A National Survey of Early-Adopting Public Housing Authorities

**DOI:** 10.3390/ijerph19073854

**Published:** 2022-03-24

**Authors:** Ellen Childs, Alan C. Geller, Daniel R. Brooks, Jessica Davine, John Kane, Robyn Keske, Jodi Anthony, Vaughan W. Rees

**Affiliations:** 1Division of Health and Environment, Abt Associates, Rockville, MD 20852, USA; ellen.childs@gmail.com; 2Department of Social and Behavioral Sciences, Harvard T.H. Chan School of Public Health, Boston, MA 02115, USA; ageller@hsph.harvard.edu (A.C.G.); jdavine@hsph.harvard.edu (J.D.); 3Department of Epidemiology, Boston University School of Public Health, Boston, MA 02118, USA; danbrook@bu.edu; 4Boston Housing Authority, Boston, MA 02111, USA; john.kane@bostonhousing.org; 5Football Players Health Study, Harvard Medical School, Boston, MA 02215, USA; robyn_keske@hms.harvard.edu; 6Mathematica, Cambridge, MA 02114, USA; jodianthony@gmail.com

**Keywords:** secondhand smoke, smoke-free housing, tobacco control, policy implementation, socioeconomic disadvantage, health disparities

## Abstract

Secondhand smoke (SHS) exposure causes chronic illness and occurs at a higher prevalence in low-income communities than the general public. In 2018, the U.S. Department of Housing and Urban Development (HUD) instituted a smoke-free housing rule for Public Housing Authorities (PHAs) to address persistent health inequities. However, the success of smoke-free housing requires evidence to inform effective implementation approaches. A mixed-methods, cross-sectional survey was conducted in a national sample of PHAs. Questions focused on housing officials’ use of specific implementation strategies. Adjusted odds ratios were used to assess associations between implementation approaches and variations among PHAs (i.e., region, size, or recency of policy adoption). Qualitative analyses were conducted to assess the perceived effectiveness of implementation strategies. Resident engagement, staff training, and smoking cessation support were the most frequently used implementation strategies. Engagement with local stakeholders was cited less frequently. Enforcement actions were limited with no violations referred to housing court. Support for policy adherence was identified as a sixth implementation strategy. While most PHAs used at least some evidence-informed implementation strategies, a lack of a systematic approach may limit overall effectiveness. Further research is required to resolve implementation barriers experienced disproportionately by a subset of PHAs, and to inform a best practice implementation framework that meets the needs of a heterogeneous population.

## 1. Introduction

Chronic secondhand smoke (SHS) exposure is associated with serious illness, especially in children, the elderly, and those with compromised immune systems [1]. Multi-unit dwellings are a major site of SHS exposure because of the capacity for emissions to travel through buildings, resulting in SHS incursion in neighboring units, including those of non-smokers [2]. In the U.S., SHS exposure occurs at a far greater prevalence among people living below the poverty level (47.9%) compared with those at or above the poverty level (21.2%) [3]. Because those who are socioeconomically disadvantaged are more likely to live in multi-unit dwellings, combined with higher smoking rates in this population compared with the general public, SHS exposure is recognized as an important driver of health disparities [3,4,5].

To address persistent disparities in SHS exposure among residents of federally-assisted public housing, the U.S. Department of Housing and Urban Development (HUD) adopted a federal rule that went into effect in 2018 that requires Public Housing Authorities (PHAs) to implement a smoke-free housing policy [6,7]. HUD’s rule requires PHAs to impose a ban on consumption of lit tobacco products in all indoor areas and within 25 feet of all federal public housing or administrative buildings. Smoke-free housing policies can lead to reduced SHS exposure, as well as increase smoking cessation efforts, and lower costs of cleaning or refurbishing units [8,9,10]. However, details on how the rule is implemented are left to the discretion of individual PHAs, creating challenges for public housing officials as they work to adhere to a federal government mandate while addressing the health and social needs of their communities. Significant barriers related to resident adherence and enforcement of smoke-free rules have been noted, underscoring the need for evidence to guide best-practice approaches to smoke-free housing policy implementation [9,11,12,13].

Policy implementation is an ongoing process that involves preparing and initiating a planned change, followed by sustained support and monitoring [14]. Implementation of a smoke-free rule required the use of defined strategies to support initial adoption and longer term policy sustainability [15]. Officials working in PHAs have described their use of practical strategies to implement smoke-free housing rules, which include: building partnerships with local public and private agencies to access technical support and resources; consulting and engaging residents in policy approaches [16]; providing smoking cessation options [11,13,17,18,19]; and clear messaging and consistency around enforcement [11,13,17,19]. However, research to date has focused on the experiences of small or regionally limited samples of properties, and the advantages of these strategies on policy implementation have not been reported [8,11,13,17,18,19,20,21,22].

This limited evidence base underscores an urgent need to understand the barriers and facilitators to successful policy implementation. Implementation of smoke-free housing is unlikely to be accomplished with a singular approach because of substantial differences across and within PHAs, which may shape when and how a given implementation strategy might be used to optimal effect. For example, PHAs may differ in terms of their geographic location (reflecting variations in smoking prevalence and social norms), the size of their resident population, their financial and logistical resources, the physical environment of a housing development, and staff readiness to support policy implementation. Resident populations may also vary—support for smoke-free rules and intentions to comply may also influence policy success. The heterogeneity of public housing properties and their residents highlights the limitations of a “one size fits all” approach to best-practice policy implementation. As a key step towards a best practice framework, we conducted the first national assessment of PHAs that had voluntarily adopted a smoke-free policy (before and including 2015) to identify the implementation strategies perceived as most effective by PHA officials charged with smoke-free policy implementation, with respect to PHA size, region, and length of time since adoption.

## 2. Materials and Methods

### 2.1. Design and Setting

We used a cross-sectional, mixed-methods design from a stratified random sample of Executive Directors (EDs) of Public Housing Authorities (or their designees) that had voluntarily adopted a smoke-free policy. We used mixed methods to provide a more comprehensive view of smoke-free housing implementation, using open-ended answers to complement and add depth to a quantitative survey. EDs were selected as the target data source because their role typically entails the day-to-day responsibility of managing the PHA, including implementing policy directives. We allowed the EDs to refer the interview request to a designated person who was familiar with the smoke-free policy rollout. Data were collected using a web-based survey between September 2016 and April 2017.

### 2.2. Survey

Demographic and personal history questions included the respondent’s role at the PHA, time working at the PHA, involvement in planning and preparation, and ongoing policy implementation. Policy details included the year the smoke-free policy was adopted, whether the policy prohibits smoking everywhere on the property or where residents are allowed to smoke, and whether e-cigarette use was allowed in indoor settings (such as private apartments).

Based on prior work, the existing literature, HUD’s published smoke-free policy implementation guidance, and conversations with stakeholders in public housing and related health agencies [23], we identified five key implementation strategies likely to impact smoke-free policy success: resident engagement, smoking cessation, staff training and support, external partnerships, and enforcement strategies. We structured questions around the planning phase (the activities leading up to the date of the smoke-free policy adoption) and the implementation phase (the activities after the date of the policy adoption). We also asked open-ended questions on the direct experiences of PHA staff in planning and implementing their smoke-free policy, and advice they would give to housing officials faced with similar challenges. A copy of the survey is available in Appendix A.

### 2.3. Sample and Recruitment

A sampling frame was developed based on a list provided by HUD of PHAs that had voluntarily adopted a smoke-free rule before September 2015. To ensure appropriate representation we stratified our sample by: (i) geographic region; (ii) PHA size (resident population); and (iii) date of adoption. Geographic region was stratified because of regional differences in smoking rates and non-smoking laws and regulations. We combined HUD’s 10 regional geographic units [24] into five broader categories (New England, Mid-Atlantic, Midwest, South, and West) and selected similar numbers of PHAs from each grouping. We stratified by date of adoption to address variations in implementation recall and experience over time. Policies take between two and four years to become fully implemented: thus, we assumed that earlier adopters have more extensive experience with policy implementation [25]. Our sample included 20% that had adopted before 2012, 70% from 2012–2014, and 10% in 2015. We stratified by PHA size because smaller PHAs may have fewer residents to engage but also fewer staff to engage them, whereas larger PHAs may have more staff and resources to support policy change but a more diffuse community. Using HUD’s administrative categories, we defined size as small (1–249 units), medium (250–1249 units), and large (1250+ units).

We identified 421 eligible PHAs from a total of 612 known early adopter PHAs. We sampled 250 PHAs using the sampling frame described above, with a goal of gaining a final sample of N = 150. We used Qualtrics to distribute unique survey links to PHA Executive Directors, and recontacted initial non-responders via email and phone.

### 2.4. Quantitative Analysis

Descriptive statistics were generated for characteristics of participating PHAs (size, year of policy adoption, and combined HUD region) and individual respondents (job type, and extent of involvement during the planning and implementation phases), and policy approaches (rules on e-cigarettes and outdoor smoking on property).

We next assessed self-reported use of the five targeted smoke-free implementation strategies according to the type and number of activities, and perceived usefulness of each strategy. We then categorized each PHA as having conducted none, few, or most/all (that is, more than half) of the listed activities for each strategy based on summation of activities cited. We identified “highly engaged” PHAs, that is those who engaged their residents and staff in the planning and preparation phase of the implementation process, based on those which undertook most or all of the listed activities related to resident engagement, smoking cessation, and staff training. Logistic regression models were generated to estimate the odds of performing most/all activities compared to few/none, with adjustment for PHA region, size, and year of adoption.

### 2.5. Qualitative Analysis

We developed a codebook using a collaborative approach [26,27]. Three research team members read selected qualitative data excerpts independently to generate codes and definitions. We discussed potential codes and developed a preliminary codebook that team members then applied independently to another set of excerpts. We iteratively compared code applications and discussed and resolved discrepancies until reaching a consensus. Final codes, which represented respondents’ perceptions on the implementation approaches they used (i.e., “perceived effectiveness”), were then applied to the open-ended responses using NVivo.

### 2.6. Mixed Methods Synthesis

We concurrently collected the quantitative and qualitative data. We used a concurrent nested approach [28] in which the quantitative data were analyzed first. We then used qualitative data analyses to confirm and provide explanatory depth to our quantitative findings. The qualitative data illuminated emergent challenges and opportunities to expand our understanding of best practices.

The research protocol was approved by the Harvard Longwood Campus and Boston University Medical Campus Institutional Review Boards (IRBs).

## 3. Results

We received complete responses from 158/250 PHAs (63% response rate). Four PHAs did not have a smoke-free policy covering 100% of indoor spaces, leaving N = 154 available for analysis. Most respondents were Executive Directors (66%), with other respondents comprising PHA staff who either served in a senior management role (e.g., Senior Administrators or Property Managers) or worked in an advocacy role on behalf of residents (Resident Service Coordinators). A great majority of respondents had worked at the PHA for ≥10 years (61.7%) and reported close involvement in the policy planning process (71.4%) and/or assisted with policy implementation after initial adoption (73.4%) (Table 1). Table 2 summarizes these PHAs by region, size, and year of adoption.

### 3.1. Policy Components

Almost all PHAs (95.4%) incorporated the smoke-free policy into a formal lease agreement. A minority (19.4%) of PHAs prohibited smoking everywhere on the property. Of the PHAs that allowed smoking on property, 26.0% had a designated outdoor smoking area, and 47.4% imposed a ban on smoking within a specified distance from PHA buildings (i.e., “buffer zone”). Almost half of PHAs allowed the use of e-cigarettes on the property (46.9%), yet just 2.0% allowed e-cigarette use in private apartments. A greater proportion of earlier-adopter PHAs allowed e-cigarette use. Table 3 reports the number of PHAs that adopted a property-wide smoking ban or allowed e-cigarette use.

### 3.2. Resident Engagement

Most PHAs hosted resident information sessions (71.3%) and surveyed residents about their support for the policy (65.8%). Most respondents considered these activities helpful (73.3%). PHAs that adopted the policy in 2015 were more likely to engage residents compared to the earliest adopters (Table 4). A minority of PHAs engaged residents for input on the policy prior to adoption, including locations where smoking is permitted on the property (44.2%), use of e-cigarettes (18.3%), and enforcement processes (36.9%). Respondents considered it helpful to consult residents about where smoking should be permitted (66.3%) and enforcement processes (60.9%). PHAs in the New England region were more likely to engage residents in policy input compared with other regions (Table 4).

In open-ended answers, respondents reiterated the importance of surveying residents and having informational meetings. Respondents reported that surveying the residents, even if the decision to go smoke-free had already been made, opened the topic for discussion and allowed the PHA staff to gauge support for the policy. As one ED reported:


*I think the initial survey started the open discussion process of the policy. Questions presented at that time were immediately addressed to put their minds at ease…I feel this gave residents a feeling of being able to share their concerns and be heard.*
—Midwest, small sized PHA, adopted 2012–2014

One respondent explained the avenues for sharing information and providing education at the meetings:


*Resident meetings were held at each site. This gave PHA staff an opportunity to present the health & safety benefits of going smoke-free. It then provided the residents to voice their opinion on the subject. Members of our Board of Commissioners were able to attend some of these meetings and hear from the residents directly.*
—New England region, medium sized PHA, pre-2012

Many PHAs found success in including residents in decision-making to increase the sense of agency and commitment to the policy. One respondent described their methods this way:


*Participation of all the residents was key. It is their home, so they got to decide “grandfathering” or not allowing any smoking…We worked really hard to NOT make the smoking residents feel like they were being “called out”. Everyone got to vote, everyone got a voice. Overall, they all decided to not allow smoking in their house.*
—Midwest, medium sized PHA, pre-2012

### 3.3. Smoking Cessation Opportunities

Most PHAs provided, at a minimum, cessation information (75.5%) or referrals to smoking cessation services (72.1%). Overall, PHAs offered a range of smoking cessation options, including telephone quitline information (79.1%), quit smoking materials (75.5%), outside referrals (72.1%), advice to contact a primary healthcare provider (41.7%), onsite counselling (36.6%), or nicotine replacement therapy (27.6%). Almost half of the PHAs (44.2%) provided most or all of the smoking cessation options listed in the survey, while 14.3% provided no resources. Large PHAs were more likely to provide a greater range of cessation options compared with medium and small PHAs (aOR = 4.76; 95% CI = 1.35, 16.70) (Table 2). Most PHAs provided cessation support in combination with education about the policy rules for residents interested in quitting smoking. EDs recommended adding cessation classes or other cessation support after policy adoption. Most PHAs provided cessation supports in combination with education about the policy specifics to assist smokers who were interested in quitting, such as this respondent:


*Added several smoke free messages to our resident calendar to remind residents of policy and cessation assistance. Several articles throughout the year prior to implementation in the resident monthly newsletter. Offered free cessation services; partner offered free [nicotine] gum, lozenges, and other products. Planned very well and had cross-disciplinary team to plan for a year and implement.*
—South region, large sized PHA, adopted 2015

### 3.4. Staff Support and Training

Most EDs provided staff training, including general information about the policy (94.0%), procedures for identifying violations (86.6%), mechanisms for enforcement or responding to violations (84.8%), identification of smoking cessation resources (66.4%), and information on the health risks of secondhand smoke exposure (58.7%). Some EDs provided training on resident outreach and engagement (50.3%), general communication and negotiation skills (47.5%), and information about smoking cessation (24.8%). Over half of PHAs trained their staff on most/all of these topics (57.8%), while 37% trained on a few of the topics. Medium and large PHAs were more likely to use most/all staff training strategies, compared with small PHAs. New England region PHAs were more likely than PHAs from other regions to employ all or most training strategies (Table 2). EDs highlighted the importance of reinforcing the policy information to staff and contractors or vendors. As one ED reported:


*Education is the key. Make sure both staff and residents understand the reason for smoke free housing. Stress that the policy is not about the smokers, but about the smoke. People become very defensive when they feel that they are being told to do something that will make them healthier, so keep it about second and third hand smoke.*
—South region, medium sized PHA, adopted 2012–2014

### 3.5. External Partnerships

Thirty-nine percent of PHAs utilized technical assistance or training provided by external organizations, and 91.7% of respondents from those PHAs considered it “helpful” or “somewhat helpful.” PHAs partnered with local health departments (49.4%), community health centers (26.0%), community service agencies (23.4%), and local chapters of NGOs such as American Lung Association (25.3%). Cessation support (22.1%) was the most common type of support received. Larger PHAs reported extensive support from external partners, from surveying residents to providing educational materials throughout the implementation process. Some PHAs reported extensive engagement with local community health agencies or tobacco control programs, from surveying the residents, to providing educational materials throughout the implementation process, for example:


*We partnered with our local [tobacco education] program to assist in survey, education, training, and implementation. This partnership was extremely helpful in educating our residents, staff, and board on the benefits of choosing a smoke free housing environment.*
—West region, medium sized PHA, adopted 2012–2014

Others used guidance from HUD or other PHAs, such as this ED:


*Be very methodical and open about the process. Keep emphasizing the positive outcomes. Use all the resources out there—other housing authorities, HUD guidance is great.*
—South region, medium sized PHA, adopted 2012–2014

### 3.6. Enforcement Activities

The most common enforcement action involved issuing a violation notice, including meeting with tenants to discuss policy violations. Some PHAs fined residents for violating the policy, and fewer reported using eviction. Similar proportions of PHAs used all (38.3%), a few (26.0%), or none (35.7%) of these strategies. PHA staff received violation reports from residents (93.5%) and maintenance staff (91.5%), with a median of 5 violations in the last year (range 0–79). PHAs seldom pursued court hearings or eviction procedures (range = 0–10).

Respondents described enforcement as the most difficult part of policy implementation. Some expressed frustration that residents continue to smoke indoors, especially after business hours. PHA staff described efforts to apply a consistent approach, as this ED stated:


*Some of [the residents] are still thinking they are clever and learn to hide it better but smoke has a raunchy odor…It makes my job harder being the enforcer. When I knock on a door, I no longer ask if they are smoking…I can smell it and start off saying “you are smoking in your apartment and that is a violation. This will be recorded in your file and three strikes you’re out”… sometimes it gets their attention, other times it doesn’t.*
—South region, small sized PHA, adopted 2012–2014

Respondents who undertook formal eviction procedures reported challenges on whether housing courts considered the smoke-free policy as grounds for eviction, and what constituted appropriate evidence.


*We have adopted our policy to make it stronger and are now able to initiate the position to fine and evict tenants that do not follow the lease. The courts did find it hard to evict a tenant that was paying rent but was smoking.*
—New England region, small sized PHA, adopted 2015

### 3.7. Adherence Support

Supporting adherence among residents who smoke emerged as a sixth implementation strategy from the qualitative analysis of open-text responses and was confirmed through subsequent interviews and focus groups [29]. PHA staff recognized the importance of being flexible and supportive of residents throughout the implementation process. PHA staff understood that it is difficult to quit smoking and worked proactively with residents to support their adherence. For example, an ED commented:


*“Be persistent but patient. We have all seniors. Our problem smokers have been smoking on average for over 45 years. They cannot and will not just stop. But in time, some will, and some will fully comply.”*
—Mid-Atlantic region, medium sized PHA, adopted 2012–2014

Approaches to supporting adherence varied. Housing staff reported that they provided smoking cessation classes, while some started the policy during warmer months to get residents accustomed to smoking outside and provided clear designated smoking areas on the property with cigarette receptacles. Although e-cigarette use might be seen as a potential strategy to help residents adhere to a smoking ban, PHAs overwhelmingly did not support indoor use of those products.

### 3.8. Highly Engaged PHAs

Summation scores revealed that some PHAs were more likely to engage residents and staff in the implementation planning phase. Large (aOR = 4.23; 95% CI = 1.26, 14.19) and medium (aOR = 3.13; 95% CI = 1.21, 8.12) PHAs were more likely to report greater resident engagement, smoking cessation support, and staff training than small PHAs. Late adopter PHAs were more likely to use more of these activities than earlier adopters (aOR = 7.13; 95% CI = 1.48, 34.26). PHAs in the South (aOR = 0.25; 95% CI = 0.08, 0.78) and West (aOR = 0.23; 95% CI = 0.06, 0.85) were less likely to engage in multiple activities than those in New England (Table 4).

Qualitative data provided further insight on these findings. EDs highlighted the advantages of engaging staff and residents across a range of implementation activities, for example:


*“Involve residents, be prepared for opposition, secure staff buy in and make it fun. Offer lots of support for residents and be prepared for procedures around enforcement.”*
—New England region, small sized PHA, adopted 2012–2014

Nonetheless, some housing officials pointed to systemic factors that prevented them from incorporating a wider range of implementation strategies, including limited capacity (personnel and financial resources) associated with being a smaller PHA and limited external resources associated with some geographic areas:


*“We live in rural area and getting cessation support is difficult and generally utilizing the internet is our only source. Our Smoking Policy has a two strikes and you’re out. You cannot sway on this policy. All smokers and non-smokers are watching to make sure you follow through with what they signed…so we do. I document everything and write violation letters.”*
—Midwest region, small sized PHA, adopted pre-2012

Despite noted challenges in adopting multiple implementation strategies, housing officials reflected a strong theme of compassion for their residents and a willingness to strive to make the policy successful for all, expressed in this quote:


*“Get the residents involved as soon as possible. Tell them you are thinking about it, ask them their opinion and suggestions on how to implement, how to help those who do smoke and will find it very difficult to quit. Go about it with compassion, understanding that smoking has been shown to help some people with certain illnesses.*
—West region, medium sized PHA, adopted pre-2012

## 4. Discussion

Surveys conducted with staff from a national sample of PHAs highlighted a range of strategies to support smoke-free policy planning and implementation, including five implementation strategies identified a priori: resident engagement, smoking cessation, staff training, external partnerships, and enforcement. Examples of each of these implementation strategies have been reported in smaller surveys of PHAs and other providers of low-income housing, although this is the first demonstration of the use of all of these five strategies within one study [11,12,17]. The present findings, when considered in the context of existing research, provide support for a set of “universal” approaches used by housing officials to implement smoke free policies. A majority of PHAs made extensive use of four implementation strategies—resident engagement, smoking cessation, staff training, and enforcement—but reported limited use of partnerships with external agencies such as health departments, health providers, and tobacco control organizations. Such agencies have been identified in previous studies as an important source of technical and logistical support [30,31]. This may reflect the challenge of adopting initiatives that depend on external resources, in preference to strategies that can be accomplished “in-house”. Nonetheless, housing officials who did develop external partnerships described their experiences in positive terms, highlighting an underdeveloped opportunity for housing providers to strengthen their implementation plans.

Evidence for a sixth implementation strategy—adherence support—was reflected in practical approaches adopted by PHA staff to accommodate needs and preferences of residents who smoke, both in transitional and longer-term policy maintenance phases. PHA staff described an approach to supporting residents in a manner akin to the principles of harm reduction: by acknowledging that a proportion of residents will continue to smoke [32], some PHAs created an opportunity to focus on ways to ensure that smoke-free rules would be followed to the advantage of all. Practical strategies such as timing the transition for the summer months (when outdoor smoking is more manageable), allowing the use of reduced emission e-cigarettes, providing nicotine replacement products to dissuade smoking indoors, and providing a designated smoking area on the property were seen as ways to support adherence among residents who were not ready to quit. E-cigarettes were seldom allowed to be used indoors but were permitted for use outdoors by about half of the PHAs: about the same proportion that allowed outdoor smoking. There was no evidence that e-cigarettes were viewed as a mechanism to support adherence, even as a transitional strategy. We also note that a greater proportion of early-adopting (pre-2012) PHAs allowed e-cigarette use, suggesting that this policy option may have become less popular among later adopting PHAs as e-cigarette use rates and product types changed dramatically since 2012. While virtually all PHAs reported at least some use of recommended implementation strategies, it was also apparent that implementation plans were developed and applied on an ad-hoc basis, often using a trial-and-error approach. This is understandable given the current absence of a comprehensive, evidence-based implementation road map to address implementation barriers. While newer resources have been made available to housing providers, including formal guidance from HUD, many PHAs we surveyed had accessed little such guidance [23]. Even so, we observed some “highly engaged” PHAs which were more likely to be large, more recent adopters of a smoke-free policy located in the New England, Midwest, and Mid-Atlantic regions.

The reasons for the observed variations in the “comprehensiveness” of implementation plans across PHAs, and capacity to overcome implementation barriers requires deeper scrutiny. Presumably, implementation strategies operate synergistically, increasing the effectiveness of an overall implementation plan beyond the contribution of the individual strategies. For example, we might speculate that enhancing resident engagement may lead to greater uptake of cessation opportunities (through communication and enhancement of trust) while decreasing the need for enforcement actions (through improved resident support for the policy and broadening of adherence options). Our evidence suggests that access to resources may be a significant barrier to employing more implementation strategies: smaller PHAs tend to lack resources and may struggle to engage staff and residents. Further research is needed to learn more about the obstacles facing smaller PHAs, which represent more than half of all PHAs, and tend to be located in non-urban areas. It is quite striking that smaller PHAs were nearly five times less likely than larger PHAs to report fewer cessation supports. Further investigation is warranted to see whether it would be both viable and beneficial for national NGOs to provide more tailored support for smaller PHAs and those with fewer resources.

Variations in social norms on smoking and the perceived fairness of smoke-free policies may also present implementation barriers for some PHAs [13,33]. The challenges faced by PHAs facing such systemic barriers, including limited technical, financial, and logistical resources, might be addressed, at least in part, by partnerships with local agencies. Feedback from PHA staff also suggests that gaining the commitment of all housing staff (including maintenance and security), and engaging residents early in the process may further reduce implementation barriers.

Nonetheless, the future success of smoke-free housing will be made more secure with rigorous evidence to guide the selection of implementation strategies to address persistent implementation barriers to ensure optimal policy impact. Successful policy implementation requires action plans that are feasible and sustainable. Using results from this project, we developed the Building Success: Implementing Effective Smoke-Free Housing Policies website (https://buildingsuccesssmokefree.org) accessed on 15 February 2022 [29] to support smoke-free housing policy implementation with the use of evidence-informed strategies and tools.

### Limitations

Several limitations are noted. Since these data were collected, HUD has instituted a federal smoke-free rule for all PHAs. However, despite the fact that PHAs are now smoke-free, ongoing challenges in policy implementation are still highly relevant, both among PHAs and in other forms of federally subsidized housing yet to adopt a smoke-free rule. Although there may be a considerable number of implementation strategies (and combinations thereof) available to PHAs, we focused our investigation on a relatively limited set of strategies. Nonetheless, the strategies we opted to explore were supported by evidence generated from previous research. We did not directly assess outcomes related to policy success, such as changes in SHS exposure or ambient air quality, nor are we able to link the reported implementation strategies with health outcomes [34]. The evidence presented here is restricted to the self-reports of PHA officials and lacks direct input from a wider range of PHA staff as well as PHA residents. These broader perspectives are critical to gain deeper insight into the complexity of smoke-free housing implementation experiences [35]. While a high proportion of PHAs reported resident engagement activities, we do not know whether those activities had high participation rates or how residents perceived them. Moreover, this is a study of voluntary adopters who may have different motivations to implement smoke-free policies compared to those under mandate. While we sampled strategically to overcome concerns of recall bias for the earliest adopters and lack of history with implementation for later adopters, we are unable to assess potential bias arising from these diverse experiences [36].

## 5. Conclusions

By describing how early-adopting PHAs implemented smoke-free policies, we provide insights on key drivers of policy implementation and how they vary across contexts that include PHA size, region, and recency of policy adoption. These findings confirm the perceived advantages of using a combination of six implementation strategies (i.e., resident engagement, staff training, smoking cessation, engagement with local stakeholders, enforcement actions, and support for policy adherence), tailored to meet the needs of individual PHAs. We also demonstrate the uneven use of these strategies across PHAs, with smaller properties, and those in the South and Western regions, less likely to adopt multiple implementation approaches owing to limited capacity or opportunity. Further work is needed to refine the six-strategy framework and provide a rigorous evidence base for specific tools and strategies to support a best practice approach to smoke-free policy implementation. These findings may also support wider adoption of smoke-free policies in other forms of multi-unit housing beyond PHAs, including the larger population of federally-subsidized, privately managed affordable housing communities [37].

## Figures and Tables

**Table 1 ijerph-19-03854-t001:** Characteristics of PHA staff respondents.

Position	N (%)
Executive Director	101 (65.6)
Senior Administrator	32 (20.8)
Property Manager	10 (6.5)
Resident Service Coordinator	7 (4.6)
Other	4 (2.6)
Duration of work at PHA	
Less than 1 year	4 (2.7)
Between 1–3 years	9 (5.8)
3–5 years	15 (9.7)
6–10 years	31 (20.1)
More than 10 years	95 (61.7)
Presence at planning and preparation phase	
No	20 (13.0)
Yes	134 (87.0)
Involvement at planning and preparation phase	
To a great extent	110 (71.4)
Somewhat	19 (12.3)
Very little	3 (1.9)
Not at all	22 (14.3)
Involvement in ongoing implementation of smoke-free policy	
To a great extent	113 (73.4)
Somewhat	28 (18.2)
Very little	10 (6.5)
Not at all	3 (2.0)

**Table 2 ijerph-19-03854-t002:** Number (and percent) of participating PHAs by Region, Size, and Year of Adoption.

	PHA Size N (%)			Year of Policy Adoption N (%)			Total by Region N (%)
PHA Region	Small	Medium	Large	Pre-2012	2012–2014	2015	
New England	15 (57.7%)	8 (30.8%)	3 (11.5%)	3 (11.5%)	22 (84.6%)	1 (3.85%)	26 (16.9)
Mid Atlantic	8 (38.1%)	11 (52.4%)	2 (9.5%)	4 (19.1%)	11 (52.4%)	6 (28.57%)	21 (13.6)
Midwest	11 (45.8%)	8 (33.3%)	5 (20.8%)	6 (25.0%)	17 (70.8%)	1 (4.17%)	24 (15.6)
South	33 (70.2%)	8 (17.0%)	6 (12.8%)	9 (19.2%)	35 (74.5%)	3 (6.38%)	47 (30.5)
West	27 (75.0%)	9 (25.0%)	0 (0.0%)	11 (30.56%)	22 (61.1%)	3 (8.33%)	36 (23.4)
Category Totals	94 (61.0)	44 (28.6)	16 (10.4)	33 (21.4)	107 (69.5)	14 (9.1)	154 (100)

**Table 3 ijerph-19-03854-t003:** Policy variations on outdoor smoking and e-cigarette use among PHAs, by region, size, and year of adoption.

	Total Property-Wide Smoking Ban	Allowed E-Cigarette Use
PHA Region		
New England	8 (30.8%)	11 (42.3)
Mid Atlantic	5 (23.8%)	9 (42.9%)
Midwest	7 (29.2%)	12 (52.1%)
South	4 (8.5%)	22 (52.3%)
West	6 (16.7%)	12 (34.3%)
PHA size		
Small	17 (18.1%)	40 (44.4%)
Medium	8 (18.2%)	17 (41.5%)
Large	5 (31.3%)	9 (56.3%)
Year of adoption		
Pre-2012	6 (18.2%)	18 (58.1%)
2012–2014	20 (18.7%)	44 (47.8%)
2015	4 (28.6%)	4 (28.6%)
Total	30 (19.4)	66 (42.9)

**Table 4 ijerph-19-03854-t004:** Adjusted odds ratios (95% CI) of undertaking most/all specified implementation strategies, by size and year of adoption and region.

	aOR (95% CI)
Residents Engaged Prior to Adoption	
Region: Mid-Atlantic (v. New England)	0.71 (0.2, 2.52)
Region: Midwest (v. New England)	0.83 (0.26, 2.71)
Region: South (v. New England)	0.35 (0.12, 1.04)
Region: West (v. New England)	0.39 (0.12, 1.29)
Size: Medium (v. small)	2.20 (0.9, 5.41)
Size: Large (v. small)	2.22 (0.68, 7.2)
Year: 2012–2014 (v. pre-2012)	2.29 (0.77, 6.87)
Year: 2015 (v. pre-2012)	**6.56 (1.48, 29.11)**
**Residents Involved in Decision Making**	
Region: Mid-Atlantic (v. New England)	**0.23 (0.06, 0.89)**
Region: Midwest (v. New England)	0.31 (0.09, 1.05)
Region: South (v. New England)	**0.28 (0.1, 0.81)**
Region: West (v. New England)	**0.17 (0.05, 0.62)**
Size: Medium (v. small)	1.87 (0.74, 4.72)
Size: Large (v. small)	2.06 (0.62, 6.85)
Year: 2012–2014 (v. pre-2012)	1.21 (0.42, 3.47)
Year: 2015 (v. pre-2012)	2.35 (0.54, 10.33)
**Cessation Support Options**	
Region: Mid-Atlantic (v. New England)	0.5 (0.14, 1.75)
Region: Midwest (v. New England)	0.67 (0.21, 2.16)
Region: South (v. New England)	0.53 (0.19, 1.46)
Region: West (v. New England)	0.47 (0.16, 1.39)
Size: Medium (v. small)	1.32 (0.59, 3.00)
Size: Large (v. small)	**4.76 (1.35, 16.73)**
Year: 2012–2014 (v. pre-2012)	0.53 (0.22, 1.28)
Year: 2015 (v. pre-2012)	2.34 (0.57, 9.64)
**Staff Training on Policy Implementation**	
Region: Mid-Atlantic (v. New England)	0.44 (0.11, 1.74)
Region: Midwest (v. New England)	0.34 (0.09, 1.27)
Region: South (v. New England)	0.41 (0.14, 1.26)
Region: West (v. New England)	0.36 (0.11, 1.15)
Size: Medium (v. small)	**3.25 (1.35, 7.83)**
Size: Large (v. small)	**19.54 (2.38, 160.7)**
Year: 2012–2014 (v. pre-2012)	0.64 (0.23, 1.61)
Year: 2015 (v. pre-2012)	1.79 (0.37, 8.71)
**Enforcement Activities**	
Region: Mid-Atlantic (v. New England)	0.98 (0.29, 3.39)
Region: Midwest (v. New England)	0.38 (0.12, 1.26)
Region: South (v. New England)	0.47 (0.17, 1.28)
Region: West (v. New England)	**0.25 (0.08, 0.8)**
Size: Medium (v. small)	2.2 (0.95, 5.1)
Size: Large (v. small)	1.98 (0.63, 6.21)
Year: 2012–2014 (v. pre-2012)	1.05 (0.42, 2.65)
Year: 2015 (v. pre-2012)	2.68 (0.66, 10.84)
**Combined implementation activities (highly engaged PHAs)**	
Region: Mid-Atlantic (v. New England)	0.35 (0.09, 1.36)
Region: Midwest (v. New England)	0.54 (0.16, 1.84)
Region: South (v. New England)	**0.25 (0.08, 0.78)**
Region: West (v. New England)	**0.23 (0.06, 0.85)**
Size: Medium (v. small)	**3.13 (1.21, 8.12)**
Size: Large (v. small)	**4.23 (1.26, 14.19)**
Year: 2012–2014 (v. pre-2012)	2.59 (0.79, 8.51)
Year: 2015 (v. pre-2012)	**7.13 (1.48, 34.26)**

Bolded values denote *p* ≤ 0.05.

## Data Availability

Data are available from the corresponding author upon request.

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
