# Peer review of "Assessing Smoke-Free Housing Implementation Approaches to Inform Best Practices: A National Survey of Early-Adopting Public Housing Authorities"

_ijerph, 2022, doi:10.3390/ijerph19073854_

Round 1
Reviewer 1 Report
This mixed methods study presents findings from surveys and interviews of early adopters of smoke-free housing policies in public housing developments across the United States. The study characterizes implementation processes across a range of PHAs, considering factors such as region and size. This study fills a gap in the literature, as most studies of multifamily SFH rules are focused on single cities or single housing authorities. Overall, the analysis is well-conducted and the manuscript is well-written.
The biggest issue I have with the manuscript is that the focus is not really on identifying “best practices” as the title and Introduction suggest. Instead it reads more like a description of implementation practices across PHAs. There is relatively little synthesis of effective strategies or guidance that other PHAs could use to improve implementation. I would suggest that the authors place more of an emphasis on synthesizing and describing which strategies are perceived to be the most effective, or reframe the title and Introduction to characterize the study as primarily descriptive in nature. For what it’s worth, the stated goals in the “Conclusions” paragraph more accurately describe the analysis than what’s written in the Introduction.
The Introduction lists many factors that may impact successful policy implementation, noting that there likely is not a one-size-fits-all approach. However, this paper only stratifies by a few of these characteristics. This is a limitation.
It would be helpful to describe the typical roles and responsibilities of a PHA ED.
“SHS exposure is recognized as an important driver of health disparities, owing to that fact that people who are socioeconomically disadvantaged are more likely to live in multi-unit dwellings, combined with higher smoking rates compared with the general public.” The phrasing of this sentence is confusing. The authors could simply cite SHS exposure statistics by poverty level, which would make their point about disparities much more clearly.
I believe that HUD actually finalized the smoke-free housing rule in 2016, not 2018. The rule required PHAs to implement their smoke-free policies by July 2018.
Typo in the sentence starting on line 59: “Officials working in PHAs have described their use of practical strategies to implement smoke-free housing rules, which include: building partnerships with local public and private agencies to; [16] consulting and engaging residents in policy approaches; [11,13,17–19] providing smoking cessation options;[11,13,17,19] and clear messaging and consistency around enforcement [11,13,17]”
Were all of the perceived effectiveness questions asked of every PHA, or only of PHAs who used a particular strategy? In other words, were PHAs who did NOT host resident information sessions asked about the perceived effectiveness of this strategy?
Table 2: can you clarify whether “Allowed e-cigarette use” captures e-cigarette use on the property (versus private apartments)?
Title of Table 3: what does “most/all” mean? Likewise, can you clarify what is meant by the subheading “Combined implementation activities (highly engaged PHAs)”?
Line 191: I’m finding it hard to reconcile how “almost half” of all PHAs offered all of the smoking cessation supports in the survey, but only 27.6% offered NRT. Please clarify.
In general, it would be helpful to further elaborate on what is meant by “most/all” strategies. This descriptor is used repeatedly in the manuscript. Does it mean more than half?
Unless I missed it, I didn’t see any presentation of the perceived effectiveness data (except in the text). I would think this data would be key to the authors’ goals.
I appreciate that the authors recognize that only surveying EDs is a considerable limitation of the analysis, as residents and other staff may have different perceptions of the implementation process. I think the authors could elaborate more on this point and on the importance of evaluating the implementation process from multiple perspectives. For example, a recent assessment of the implementation of a SFH policy within NYC Housing Authority buildings highlighted many issues not discussed in this analysis.
Jiang, N.; Gill, E.; Thorpe, L.E.; Rogers, E.S.; de Leon, C.; Anastasiou, E.; Kaplan, S.A.; Shelley, D. Implementing the Federal Smoke-Free Public Housing Policy in New York City: Understanding Challenges and Opportunities for Improving Policy Impact. Int. J. Environ. Res. Public Health 2021, 18, 12565. https://doi.org/10.3390/ijerph182312565
Author Response
We provide a detailed response to Reviewer 1's critique in the attached document.

Reviewer 2 Report
Thank you for the opportunity to review this manuscript. Overall, this paper addresses a critically-important topic (best practices for implementing smokefree housing policies among public housing authorities) however, as written, the paper is difficult for readers to understand and place in the current literature. With significant edits, the paper could be improved.
One of the largest weaknesses of the paper in its present form is that it frames the study and resulting manuscript as “mixed methods” but does not justify or explain this decision or report on any of the features of mixed methods research. I would highly recommend the authors read O’Cathain and colleagues’ “Good Reporting of A Mixed Methods Study (GRAMMS)” checklist and edit this paper to ensure these guidelines are followed in the reporting of this study. It’s unclear why a mixed methods research design was selected to help the authors answer their research question. Currently, it’s difficult to assess the true “mixing” of the methods – it sounds like the authors had a survey with a few open-ended questions. It would be helpful to readers if the authors included their instrument (or at least a more thorough list of constructs) as an appendix or supplemental material.
There’s some lack of clarity about who was responding to the survey – the authors say the survey links were sent to “PHA Executive Directors” but then only 66% of respondents were Executive Directors – who are the other 34% of respondents? People that the Executive Director forwarded the link to? More clarity here would help contextualize findings. I see that there is a bit more information in the supplemental table, however, this could use some type of brief explanation in the text. Since the only outcomes measured are PHA representative's self-responses to the survey it's really important to understand who was responding.
The inclusion/exclusion of e-cigarettes in PHA smokefree policies could use more explanation – the authors assert “a greater proportion of earlier-adopter PHAs allowed e-cigarette use,” but is it the case that these policies explicitly allow e-cigarettes or rather that policies that were written and implemented pre-2012 didn’t explicitly include e-cigarettes because the products, usage and landscape were so different than they are now?
The layout of the tables is a little confusing – it would be helpful to ensure it’s easy to pick out levels of the same variable (e.g., medium vs small and large vs small are both related to the size of the PHA). The use of the summation score could use more justification – the authors developed constructs they thought were important, developed an instrument to assess these constructs and then added scores for each construct together to show that PHAs who had higher scores had certain features in common. As-is it’s hard to understand what the reader should take away here.
Overall, I believe this paper has potential – the work is important and interesting and the survey and analyses appear to be solid. However, the paper needs significant edits to be a clear contribution to the field and easy for readers to contextualize in the tobacco control and mixed methods literature.
Citations:
O'Cathain A, Murphy E, Nicholl J. The quality of mixed methods studies in health services research. J Health Serv Res Policy. 2008;13: 92-98.
Author Response
We provide a detailed response to Reviewer 2's critique in the attached document

Round 2
Reviewer 1 Report
I believe the manuscript is considerably improved after incorporating revisions and would recommend it for publication.
Author Response
We than the Reviewer for their careful appraisal of this paper.
Reviewer 2 Report
Overall, the paper is improved. I believe the authors’ description of the mixed methods design of the project is not consistent with what is described. A sequential explanatory approach requires that the collection and analysis of quantitative data must proceed the collection and analysis of qualitative data. In sequential explanatory designs, the quantitative data analyses helps shape the qualitative data collection. Since all data (qualitative and quantitative) was collected simultaneously in this study, it feels like a sequential explanatory design is inaccurate. I’d encourage the authors to look into concurrent transformative design or an concurrent embedded design to best describe the methods they undertook.
Author Response
Again, we are grateful for the Reviewer's incisive comments and constructive feedback. We agree that the work we have reported would be more accurately described as a "concurrent nested" rather than "sequential explanatory" design. We have made this change at Line 157. We have also modified the supporting citation.
(As a brief explanation for our apparent confusion: we undertook a second phase of qualitative work, not reported here, which involved interviews with staff and residents at site visits. Thus our original plan was indeed to generate a sequential explanatory analysis, but we now plan to publish the other set of qualitative data separately.)